# Efficient Bayesian Sampling Using Normalizing Flows to Assist Markov Chain Monte Carlo Methods

**Marylou Gabrié** [1][2]   **Grant M. Rotskof** [3]   **Eric Vanden-Eijnden** [4]

## Abstract

Normalizing flows can generate complex target distributions and thus show promise in many applications in Bayesian statistics as an alternative or complement to MCMC for sampling posteriors. Since no data set from the target posterior distribution is available beforehand, the flow is typically trained using the reverse Kullback-Leibler (KL) divergence that only requires samples from a base distribution. This strategy may perform poorly when the posterior is complicated and hard to sample with an untrained normalizing flow. Here we explore a distinct training strategy, using the direct KL divergence as loss, in which samples from the posterior are generated by (i) assisting a local MCMC algorithm on the posterior with a normalizing flow to accelerate its mixing rate and (ii) using the data generated this way to train the flow. The method only requires a limited amount of *a priori* input about the posterior, and can be used to estimate the evidence required for model validation, as we illustrate on examples.

## 1. Introduction

Given a model with continuous parameters $\theta \in \Theta \subseteq \mathbb{R}^d$, a prior on these parameters in the form of a probability density function $\rho_o(\theta)$, and a set of observational data $D$ giving the likelihood $L(\theta)$ for the model parameter $\theta$, Bayes formula asserts that the posterior distribution of the parameters has probability density

$$\rho_*(\theta) = \rho(\theta|D)\rho_o(\theta) = Z_*^{-1}L(\theta)\rho_o(\theta) \qquad (1)$$

where the normalization factor $Z_* = \int_\Theta L(\theta)\rho_o(\theta)d\theta$ is the unknown evidence. A primary aim of Bayesian inference is

to sample this posterior to identify which parameters best explain the data given the model. In addition one is typically interested in estimating $Z_*$ since it allows for model validation, comparison, and selection.

Markov Chain Monte Carlo (MCMC) algorithms (Liu, 2008) are nowadays the methods of choice to sample complex posterior distributions. MCMC methods generate a sequence of configurations over which the time average of any suitable observable converges towards its ensemble average over some target distribution, here the posterior. This is achieved by proposing new samples from a proposal density that is easy to sample, then accepting or rejecting them using a criterion that guarantees that the transition kernel of the chain is in detailed balance with respect to the posterior density: a popular choice is Metropolis-Hastings criterion.

MCMC methods, however, suffer from two problems. First, mixing may be slow when the posterior density $\rho_*$ is multimodal, which can occur when the likelihood is non-log-concave (Fong et al., 2019). This is because proposal distributions using local dynamics like the popular Metropolis adjusted Langevin algorithm (MALA) (Roberts & Tweedie, 1996) are inefficient at making the chain transition from one mode to another, whereas uninformed non-local proposal distributions lead to high rejection rates. The second issue with MCMC algorithms is that they provide no efficient way to estimate the evidence $Z_*$: to this end, they need to be combined with other techniques such as thermodynamic integration or replica exchange, or traded for other techniques such as annealed importance sampling (Neal, 2001), nested sampling (Skilling, 2006), or the descent/ascent nonequilibrium estimator proposed in (Rotskoff & Vanden-Eijnden, 2019) and recently explored in (Thin et al., 2021).

Here, we employ a *data-driven* approach to aid designing a fast-mixing transition kernel along the lines of mode-hoping algorithms proposed by (Tjelmeland & Hegstad, 2001; Andricioaei et al., 2001; Sminchisescu & Welling, 2017) and learning based algorithms of (Levy et al., 2018; Titsias, 2017; Song et al., 2017). Normalizing flows (Tabak & Vanden-Eijnden, 2010; Tabak & Turner, 2013; Papamakarios et al., 2021) are especially promising in this context: these maps can approximate the posterior density $\rho_*$ as the pushforward of a simple base density $\rho_B$ (e.g. the prior

[1]Flatiron Institute, New York, NY 10010 [2]Center for Data Science, New York University, New York, NY 10011 [3]Dept. of Chemistry, Stanford University, Stanford, CA 94305 [4]Courant Institute, New York University, New York, NY 10012. Correspondence to: Marylou Gabrié <mgabrie@nyu.edu>.

Third workshop on *Invertible Neural Networks, Normalizing Flows, and Explicit Likelihood Models* (ICML 2021). Copyright 2021 by the author(s).

density $\rho_o$) by an invertible map $T : \Theta \to \Theta$. Their use for Bayesian inference, as opposed to density estimation (Song et al., 2017; 2021), was first advocated in Rezende & Mohamed (2015). Since a representative training set of samples from the posterior density is typically unavailable beforehand, these authors proposed to use the reverse Kullback-Leibler (KL) divergence of the posterior $\rho_*$ from the push forward of the base $\rho_B$, since this divergence can be expressed as an expectation over samples generated from $\rho_B$, consistent with the variational inference framework (Jordan et al., 1998; Blei et al., 2017). This procedure has the potential drawback that training the map is hard if the posterior differs significantly from the initial pushforward (Hartnett & Mohseni, 2020), as it may lead to "mode collapse." Annealing of $\rho_*$ during training was shown to reduce this issue (Wu et al., 2019; Nicoli et al., 2020).

Building on works using normalizing flows with MCMC (Albergo et al., 2019; Noé et al., 2019; Gabrié et al., 2021) here we explore an alternative strategy that blends sampling and learning where we (i) assist a MCMC algorithm with a normalizing flow to accelerate mixing and (ii) use the generated data to train the flow on the direct KL divergence.

## 2. Posterior Sampling and Model Validation with Normalizing Flows

A normalizing flow (NF) is an invertible map $T$ that pushes forward a simple base density $\rho_B$ (typically a Gaussian with unit variance, though we could also take $\rho_B = \rho_o$) towards a target distribution, here the posterior density $\rho_*$. An ideal map $T_*$ (with inverse $\bar{T}_*$) is such that if $\theta_B$ is drawn from $\rho_B$ then $T_*(\theta_B)$ is a sample from $\rho_*$. Of course, in practice, we have no access to this exact $T_*$, but if we have an approximation $T$ of $T_*$, it still assists sampling $\rho_*$. Denote by $\hat{\rho}$ the push-forward of $\rho_B$ under the map $T$,

$$\hat{\rho}(\theta) = \rho_B(\bar{T}(\theta)) \det \left| \nabla_\theta \bar{T} \right|. \tag{2}$$

As long as $\hat{\rho}$ and $\rho_*$ are either both positive or both zero at any point $\theta \in \Theta$, we can use a Metropolis-Hasting MCMC algorithm to sample from $\rho_*$ using $\hat{\rho}$ as a transition kernel: a proposed configuration $\theta' = T(\theta_B)$ from a given configuration $\theta$ is accepted with probability

$$\mathrm{acc}(\theta, \theta') = \min \left[ 1, \frac{\hat{\rho}(\theta)\rho_*(\theta')}{\rho_*(\theta)\hat{\rho}(\theta')} \right]. \tag{3}$$

This procedure is equivalent to using the transition kernel

$$\pi_T(\theta, \theta') = \mathrm{acc}(\theta, \theta')\hat{\rho}(\theta') + \left(1 - r(\theta)\right)\delta(\theta - \theta') \tag{4}$$

where $r(\theta) = \int_\Theta \mathrm{acc}(\theta, \theta')\hat{\rho}(\theta')d\theta'$. Since $\pi_T(\theta, \theta')$ is irreducible and aperiodic under the aforementioned conditions on $\rho_*$ and $\hat{\rho}$, its associated chain is ergodic with respect

---

**Algorithm 1** Concurrent MCMC sampling and map training

1: SAMPLETRAIN($U_*$, $T$, $\{\theta_i(0)\}_{i=1}^n$, $\tau$, $k_{\max}$, $k_{\mathrm{Lang}}$, $\epsilon$)
2: **Inputs:** $U_*$ target potential, $T$ initial map, $\{\theta_i(0)\}_{i=1}^n$ initial chains, $\tau > 0$ time step, $k_{\max} \in \mathbb{N}$ total duration, $k_{\mathrm{Lang}} \in \mathbb{N}$ number of Langevin steps per NF resampling step, $\epsilon > 0$ map training time step
3: $k = 0$
4: **while** $k < k_{\max}$ **do**
5:   **for** $i = 1, \ldots, n$ **do**
6:     **if** $k \mod k_{\mathrm{Lang}} + 1 = 0$ **then**
7:       $\theta'_{B,i} \sim \rho_B$
8:       $\theta'_i = T(\theta'_{B,i})$ {push-forward via $T$}
9:       $\theta_i(k+1) = \theta'_i$ with prob $\mathrm{acc}(\theta_i(k), \theta'_i)$, otherwise $\theta_i(k+1) = \theta_i(k)$ {resampling step}
10:     **else**
11:       $\theta'_i = \theta_i(k) - \tau\nabla U_*(\theta_i(k)) + \sqrt{2\tau}\,\eta_i$ with $\eta_i \sim \mathcal{N}(0_d, I_d)$ {discretized Langevin step}
12:       $\theta_i(k+1) = \theta'_i$ with MALA acceptance prob or ULA, otherwise $\theta_i(k+1) = \theta_i(k)$
13:   $k \leftarrow k + 1$
14:   $\mathcal{L}[T] = -\frac{1}{n}\sum_{i=1}^n \log \hat{\rho}(\theta_i(k+1))$
15:   $T \leftarrow T - \epsilon\nabla\mathcal{L}[T]$ {Update the map}
16: **return:** $\{\theta_i(k)\}_{k=0,i=1}^{k_{\max},n}$, $T$

---

to $\rho_*$ (Meyn & Tweedie, 2012). In addition the evidence is given by

$$Z_* = \mathbb{E}_{\rho_B} \left[ \frac{L(T(\theta_B))\rho_o(T(\theta_B))}{\hat{\rho}(T(\theta_B))} \right]. \tag{5}$$

For the scheme to be efficient, two conditions are required. First the parametrization of the map $T$ must allow for easy evaluation of the density $\hat{\rho}$ which requires easily estimable Jacobian determinants and inverses. This issue has been one of the main foci in the normalizing flow literature (Papamakarios et al., 2021) and is for instance solved using coupling layers (Dinh et al., 2015; 2017). Second, as shown by formula (3) the proposal density $\hat{\rho}$ must produce samples with statistical weights comparable to the posterior density $\rho_*$ to ensure appreciable acceptance rates. This requires training the map $T$ to resemble the optimal $T_*$.

## 3. Compounding Local MCMCs and Generative Sampling

In Algorithm 1, we present a concurrent sampling/training strategy that synergistically uses $T$ to improve sampling from $\rho_*$ and samples obtained from $\rho_*$ to train $T$. Let us describe the different components of the scheme.

**Sampling.** The sampling component of Algorithm 1 alternates between steps of the Metropolis-Hasting procedure using a NF as discussed in Section 2 and steps of a local

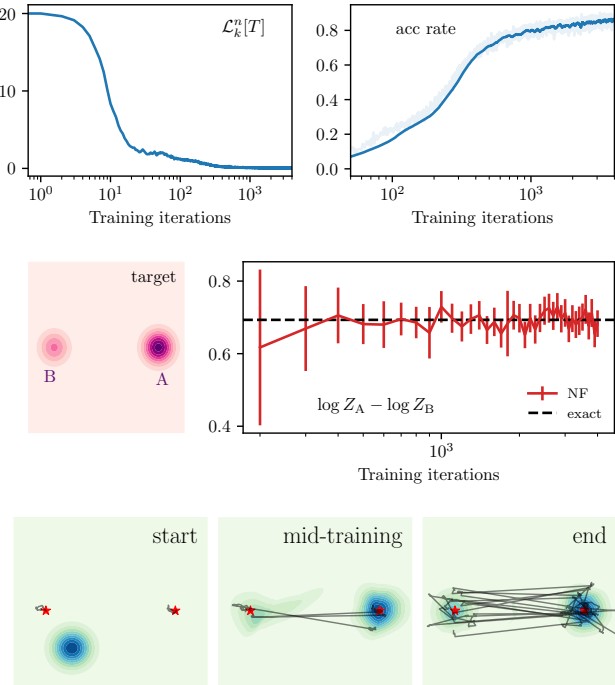

Figure 1. **Sampling a mixture of 2 Gaussians in 10d.** *Top row:* Training loss and acceptance rate of the NF non-local moves as a function of iterations. *Middle row:* Target density and estimation of the relative weight of modes A and B using sampling with $\hat{\rho}$. *Bottom row:* $\hat{\rho}$ and example chains along training/sampling. (See Appendix A.2 for setup details.)

MCMC sampler (here MALA) (line 11), using as potential

$$U_*(\theta) = -\log L(\theta) - \log \rho_o(\theta). \qquad (6)$$

Strictly speaking, the second transition kernel does not need to be local, it should, however, have satisfactory acceptance rates early in the training procedure to provide initial data to start up the optimization of $T$. From a random initialization $T$, the parametrized density $\hat{\rho}$ has initially little overlap with the posterior $\rho_*$ and the moves proposed by the NF have a high probability to be rejected. However, thanks to the data generated by the local sampler, the training of $T$ can be initiated. As training goes on, more and more moves proposed by the NF can be accepted. It is crucial to notice that these moves, generated by pushing forward independent draws from the base distribution $\rho_B(\theta)$, are non-local and easily mix between modes.

**Training.** A standard way of training $T$ is to minimize the Kullback-Leibler divergence from $\hat{\rho}$ to $\rho_*$. Since we do not have access to samples from $\rho_*$, here we use instead

$$D_{KL}(\rho_k \| \hat{\rho}) = C_k - \int_\Theta \log \hat{\rho}(\theta) \rho_k(\theta) d\theta, \qquad (7)$$

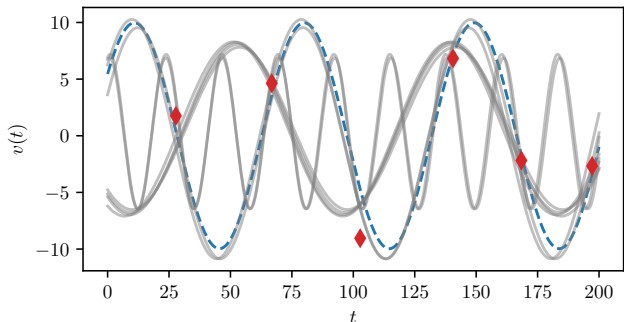

Figure 2. **Radial velocities.** From the signal plotted in blue, we draw the noisy observations in red and obtain the initial samples in gray with the Joker algorithm (Price-Whelan et al., 2017).

where $\rho_k$ denotes the density of the MCMC after $k \in \mathbb{N}$ steps and $C_k = \int_\Theta \log \rho_k(\theta) \rho_k(\theta) d\theta$ is a constant irrelevant for the optimization of $T$. In practice, we run $n$ walkers in parallel in the chain: denoting their positions at iteration $k$ by $\{\theta_i(k)\}_{i=1}^n$, we use the following estimator of (7) (minus the constant) as objective for training:

$$\mathcal{L}_k^n[T] = -\frac{1}{n} \sum_{i=1}^n \log \hat{\rho}(\theta_i(k)). \qquad (8)$$

The training component of Algorithm 1 uses stochastic gradient descent on this loss function to update the parameters of the normalizing flow (line 15). Note that $\{\theta_i(k)\}_{i=1}^n$ are not perfect samples from $\rho_*$ to start with, but their quality increases with the number of iterations of the MCMC. Note also that this training strategy is different from the one proposed in (Rezende & Mohamed, 2015), which uses instead the *reverse* KL divergence of $\hat{\rho}$ from $\rho_*$: this objective could be combined with the one in (8). Here we will stick to using (8) as loss, using approximate input from $\rho_*$ through the MCMC samples initialized as explained next. Müller et al. (2019) used the same forward KL divergence for training yet using a reweighing of samples from $\hat{\rho}$ for its estimation, which may be imprecise if the initial pushforward density bears little overlap with the target $\rho_*$.

**Initialization.** To initialize MCMC chains, we assume that we have initial data lying in each of the important modes of the posterior $\rho_*$, but require no additional information. That is, we take $\theta_i(0) = \theta_i$, where the $\theta_i$ are initially located in these modes but not necessarily drawn from $\rho_*$. We stress that the method therefore applies in situations where these modes have been located beforehand, for example by doing gradient descent on $L(\theta)\rho_o(\theta)$ from random points in $\Theta$.

Architecture details of Algorithm 1 are deferred to Appendix A, and questions of convergence to Appendix B.

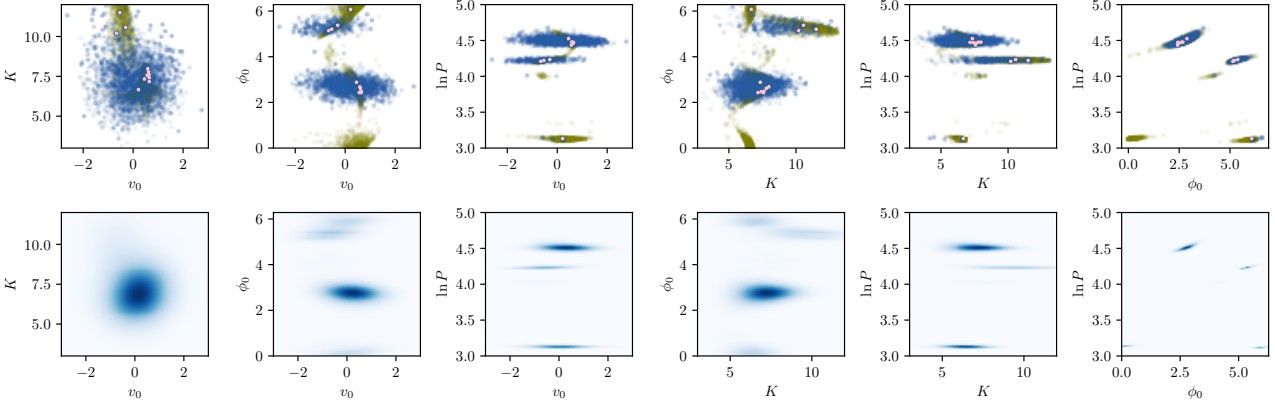

*Figure 3.* **Comparing samples from Algorithm 1** (top row in blue, 2d projections) with the samples from Joker (in green) and the initialization samples (in pink). The posterior densities marginalized over two parameters are shown in the bottom row.

# 4. Numerical Experiments

## 4.1. Sampling Mixture of Gaussians in High-dimension

As a first test case of Algorithm 1, we sample a Gaussian mixture with 2 components in 10 dimensions and estimate the relative statistical weights of its two modes.[1] The bottom row of Fig. 1 shows 2d projections of the trajectories of representative chains (in black) from initializations in each of the modes (red stars) as the NF learns to model the target density (blue contours). Running first locally under the Langevin sampler, the chains progressively mix between modes and grasp the difference of statistical weights also captured by the final map $T$. Quantitatively, the acceptance rate of moves proposed by the NF reaches $\sim 80\%$ at the end of training (Fig. 1 top row). The estimator of the relative statistical weights of each modes (the right mode A is twice as likely as the left mode B) using Eq. (5) also converges to the exact value within a small statistical error (Fig. 1 middle row).

## 4.2. Radial Velocity of an Exoplanet

Next we apply our method to the Bayesian sampling of radial velocity parameters in a model close to the one studied by Price-Whelan et al. (2017) for a star-exoplanet system. The model has 4 parameters: an offset velocity $v_0$, an amplitude $K$, a period $P$ and a phase $\phi_0$. Introducing $\theta = (v_0, K, \phi_0, \ln P) \in \Theta \subset \mathbb{R}^4$, the radial velocity is

$$v(t; \theta) = v_0 + K \cos(\Omega t + \phi_0) \quad (9)$$

with $\Omega = 2\pi/P$. From a set of observations $D = \{v_k, t_k\}_{k=1}^N$, the goal is to sample the posterior distribution

over $\theta$. Following (Price-Whelan et al., 2017), we assume a Gaussian likelihood $L(\theta) = \mathcal{N}(v_k; v(t_k; \theta), \sigma_{\text{obs}}^2)$, with known variance $\sigma_{\text{obs}}^2$, and the prior distributions

$$
\begin{aligned}
\ln P &\sim \mathcal{U}(\ln P_{\min}, \ln P_{\max}), & \phi_0 &\sim \mathcal{U}(0, 2\pi), \\
K &\sim \mathcal{N}(\mu_K, \sigma_K^2), & v_0 &\sim \mathcal{N}(0, \sigma_{v_0}^2).
\end{aligned}
\quad (10)
$$

We sample $N = 6$ noisy observations at different times $t_k$ (red diamonds in Fig. 2) from a ground-truth radial velocity with parameters $\theta_0$ (dashed blue line). Using one iteration of the accept-reject Joker algorithm (Price-Whelan et al., 2017) with $10^3$ samples from the prior distributions we obtain 11 sets of likely parameters (corresponding to the gray lines in Fig. 2), which we will use as starting points for the MCMC chains in Algorithm 1. Note that to ensure a minimum acceptance rate of $\sim 1\%$ the Joker samples priors of $P$ and $\phi_0$ only, and computes the maximum likelihood value for the "linear parameters" $K$ and $v_0$.

We assess the quality of sampling after $10^4$ iterations of Algorithm 1 on Fig. 3, looking at all the possible 2d projections of space of parameters. The samples from Algorithm 1 (top row, in blue, final acceptance rate of $60\%$, see Fig. 4) are generally covering well the modes of the marginal posterior distribution (second row), far beyond the initial samples (top row, in pink), at the exception of a light mode in the neighborhood of $v_0 = 0$ and $\phi_0 = 0$ in which no chain was initialized. This is an illustration of the need for prior knowledge of the rough location of a basin of interest to successful sample it with the proposed method. For comparison, we also report samples accepted by the Joker algorithm from an initial draw of $10^6$ samples (top row, in green). Because of the maximum likelihood step along $K$ and $v_0$ mentioned above, the posterior is not appropriately sampled along these two dimensions by this strategy.

---

[1]In this experiment and the one that follows we used unadjusted Langevin dynamics (ULA) rather than MALA because the time steps were sufficiently small to ensure a high acceptance rate.

# 5. Conclusion

Our results show that normalizing flows can be exploited to augment MCMC schemes used in Bayesian inference with a nonlocal transport component that significantly enhances mixing. By design, the method blends MCMC with an optimization component similar to that used in variational inference (Blei et al., 2017), and it would be interesting to investigate further how both approaches compare on challenging applications with complex posteriors on high dimensional parameter spaces.

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

# A. Computational details

All codes are made available through the Github repository `anonymous` (to be disclosed after double-blind review).

## A.1. Architectures

We parametrize the map $T$ as a RealNVP (Dinh et al., 2017), for which inverse and Jacobian determinant can be computed efficiently. Its building block is an invertible affine coupling layer updating half of the variables,

$$\theta_{1:d/2}^{(k+1)} = e^{s\left(\theta_{d/2:d}^{(k)}\right)} \odot \theta_{1:d/2}^{(k)} + t\left(\theta_{d/2:d}^{(k)}\right) \tag{11}$$

where $s(\cdot)$ and $t(\cdot)$ are learnable neural networks from $\mathbb{R}^{d/2} \to \mathbb{R}^{d/2}$, $\odot$ denotes component-wise multiplication (Hadamard product), and $k$ indexes the depth of the network. In our experiments we use fully connected networks with depth 3, hidden layers of width 100 with ReLU activations. The parameters of these networks are initialized with random Gaussian values of small variance so that the RealNVP implements a map $T$ close to the identity at initialization.

## A.2. Mixture of Gaussians experiment

**Target and hyperparameters.** The target distribution is a mixture of Gaussian in dimension $d = 10$ with two components:

$$\rho_*(\theta) = w_A \frac{e^{-\frac{1}{2}|\theta-\theta_A|^2}}{(2\pi)^{d/2}} + w_B \frac{e^{-\frac{1}{2}|\theta-\theta_B|^2}}{(2\pi)^{d/2}}, \tag{12}$$

with weights $w_A = 2/3$, $w_B = 1/3$ and centroids with non-zero coordinates only in the first two dimensions $\theta_{A\,1,2} = (8,3)$ and $\theta_{B\,1,2} = (-2,3)$. Fig. 1 displays density slices in the $(\theta_1, \theta_2)$-plane.

We use a RealNVP network with 6 pairs of coupling layers and a standard normal distribution as a base $\rho_B$. A set of 100 independent walkers were initialized in equal shares in modes A and B of the target. For the optimization, we compute gradients using batches of 1000 samples corresponding to 10 consecutive repeated sampling steps of Algorithm 1 (with $\tau = 0.005$ and $k_{\text{Lang}} = 1$). We run 4000 parameters updates using Adam with a learning rate of 0.005.

**Computing log-evidence differences.** Interpreting $\rho_*(\theta)$ as a posterior distribution over a set of parameters $\Theta$, we would like to estimate the relative evidence for modes $A$ and $B$. Denoting by $A$ and $B$ two sets of configurations in $\Theta$ corresponding to each mode, the difference between their log-evidence is given by

$$\log Z_A - \log Z_B = \log \frac{\int_\Theta \mathbb{1}_A(\theta)\rho_*(\theta)d\theta}{\int_\Theta \mathbb{1}_B(\theta)\rho_*(\theta)d\theta} = \log \mathbb{E}_*(\mathbb{1}_A) - \log \mathbb{E}_*(\mathbb{1}_B). \tag{13}$$

Once the normalizing flow has been learned to assist sampling, it can be used to approximate Eq. 5. Drawing $\{\theta_i\}_{i=1}^n$ from $\hat{\rho}$ we have the Monte Carlo estimate $\hat{Z}_{*,A}$ of $\mathbb{E}_*(\mathbb{1}_A)$:

$$\hat{Z}_{*,A} = \frac{1}{Z_*} \sum_{i=1}^n \mathbb{1}_A(\theta_i)\hat{w}(\theta_i), \tag{14}$$

with the unnormalized weights $\hat{w}_i = L(\theta_i)\rho_o(\theta_i)/\hat{\rho}(\theta_i)$, taking the form of an importance sampling estimator. The quality of the estimator can be monitored using an estimate of the effective sample size to adjust the variance estimate from the empirical variance

$$n_{\text{eff}} = \frac{\left(\sum_{i=1}^n \hat{w}_i\right)^2}{\sum_{i=1}^n \hat{w}_i^2}. \tag{15}$$

Finally the unknown $Z_*$ cancels out in the log-evidence difference:

$$\log Z_A - \log Z_B \approx \log \left(\sum_{i=1}^n \mathbb{1}_A(\theta_i)\hat{w}_i\right) - \log \left(\sum_{i=1}^n \mathbb{1}_B(\theta_i)\hat{w}_i\right). \tag{16}$$

In the middle right panel of Fig. 1 we report the performance of this estimator using the sets $A = \{\theta : \|\theta - \theta_A\|_2 \le 5\}$ and $B = \{\theta : \|\theta - \theta_B\|_2 \le 5\}$, and $n = 10^5$. The estimator convergence to the exact value $\ln 2/3 - \ln 1/3 = \ln 2$ as the quality of $\hat{\rho}$ increases over training.

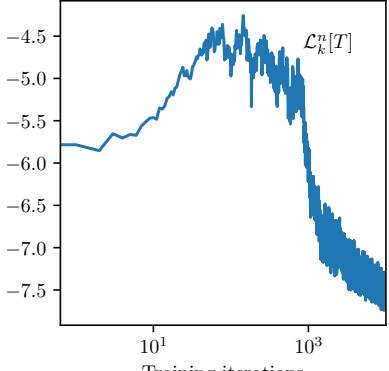 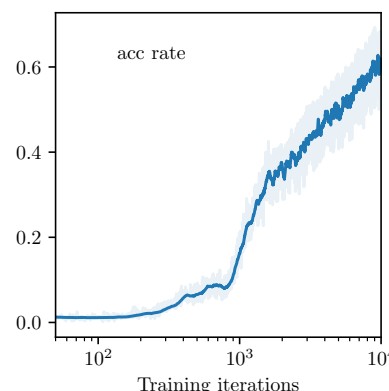 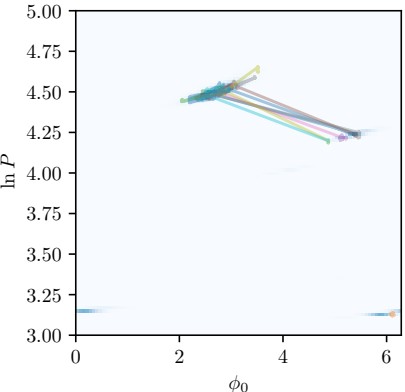

*Figure 4.* **Training of a NF to model the posterior distribution for the radial velocity experiment.** *Left:* Evolution of the approximate KL divergence used as an objective for training. *Middle:* The acceptance rate of the non-local moves proposed by the NF along training are above $60\%$ by the end of training. *Right:* The contour blue plot reports the marginalized true posterior along $\ln P$ and $\phi_0$ computed with numerical integration. The colored lines mark the trajectories of MCMC chains using the assisted sampling startegy considered here with the final learned map $T$.

## A.3. Radial velocity experiment

**Setup.** The parameters of the prior are set to the values $\ln P_{\min} = 3$, $\ln P_{\max} = 5$, $\sigma_{\mathrm{obs}} = 1.8$, $\sigma_K = 3$, $\mu_K = 5$ and $\sigma_{v_0} = 1$. The RealNVP used to learn the posterior distribution has 6 pairs of coupling layers. The base distribution $\rho_{\mathrm{B}}$ is a standard normal distribution as in the previous experiment, but the NF is learned on a whitened representation of the parameters $\theta$. Concretely, using the 11 initial samples from the Joker, we compute the empirical mean $\hat{\theta}$ and empirical covariance $\hat{\Sigma} = \sum_{i=1}^{n} (\theta_i - \hat{\theta})(\theta_i - \hat{\theta})^\top / n$. The latter admits an eigenvalue decomposition $\hat{\Sigma} = ODO^\top$, with $D$ the diagonal matrix of eigenvalues and $O$ the orthogonal eigenvectors basis. Using $W = OD^{-1/2}O^\top$, we train the normalize flow to model the distribution of the whitened parameters $\theta_W = W(\theta - \hat{\theta})$.

**Training.** A set of 110 independent walkers were initialized using 10 copies of each of 11 set of parameters retained from an initial iteration of the Joker algorithm from $10^3$ draws of $\phi_0$ and $\ln P$. Here again we use 5 consecutive repeated sampling steps of Algorithm 1 (with $\tau = 5e^{-6}$ and $k_{\mathrm{Lang}} = 1$) to compute gradients using batches of 550 samples. We run Adam for $10^4$ iterations with a learning rate of 0.001.

Fig. 4 reports the evolution of the approximate KL divergence $\mathcal{L}_k^n$ along training (left) and the acceptance rates of the non-local moves proposed by the NF (middle). We also plot the projected trajectories of 11 chains started with the initial Joker samples updated with combined sampling component of Algorithm 1 using the final learned $T$. During the short chains that included 10 non-local resampling steps, the walkers can be seen to jump back and forth between two of the modes of the marginalized density (underlying in blue in the plot). However mixing is not well assisted with the mode around $\phi_0$ (see discussion in the main text as well).

## B. Continuous limit of the MCMC scheme

### B.1. Chapman-Kolmogorov equation

Written in terms of the densities $\rho_*$ and $\hat{\rho}$ (assumed to be fixed for now) the transition kernel in (4) reads

$$\pi_T(\theta, \theta') = a(\theta, \theta')\hat{\rho}(\theta') + (1 - b(\theta))\delta(\theta - \theta') \tag{17}$$

where

$$a(\theta, \theta') = \min\left(\frac{\hat{\rho}(\theta)\rho_*(\theta')}{\hat{\rho}(\theta')\rho_*(\theta)}, 1\right),$$

$$b(\theta) = \int_\Theta a(\theta, \theta')\hat{\rho}(\theta')d\theta'. \tag{18}$$

Denoting as $\{\rho_k(\theta)\}_{k\in\mathbb{N}}$ the updated probability density of the walker in the Markov chain associated with the kernel $\pi_T(\theta, \theta')$ alone, this density satisfies the Chapman-Kolmogorov equation

$$\rho_{k+1}(\theta) = \int_\Theta \rho_k(\theta')\pi_T(\theta', \theta)d\theta'. \tag{19}$$

Using the explicit form of $\pi_T(\theta, \theta')$ in (17), after some simple reorganization this equation can be written as

$$\rho_{k+1}(\theta) = \rho_k(\theta) + \int_\Theta R(\theta, \theta')\left(\rho_*(\theta)\rho_k(\theta') - \rho_k(\theta)\rho_*(\theta')\right)d\theta' \tag{20}$$

where we defined

$$R(\theta, \theta') = R(\theta', \theta) = \min\left(\frac{\hat{\rho}(\theta)}{\rho_*(\theta)}, \frac{\hat{\rho}(\theta)}{\rho_*(\theta)}\right). \tag{21}$$

Note that if we had $\hat{\rho} = \rho_*$, then $R(\theta, \theta') = 1$ and (20) would reach equilibrium in one step, $\rho_{k+1} = \rho_*$ whatever $\rho_k$.

## B.2. Continuous limit

To take the continuous limit of (20), we modify this equation in a way that the update of the density is only partial. Specifically, denoting $\rho_t$ the value of the density at time $t \geq 0$, we turn this equation into

$$\rho_{t+\tau}(\theta) = \rho_t(\theta) + \alpha\tau \int_\Theta R(\theta, \theta')\left(\rho_*(\theta)\rho_t(\theta') - \rho_t(\theta)\rho_*(\theta')\right)d\theta' \tag{22}$$

where $\alpha > 0$ and $\tau > 0$ are parameters. This will allow us to make the MCMC resampling updates on par with those of MALA, using $\tau > 0$ as timestep in both. Subtracting $\rho_t(\theta)$ from both sides of (22), dividing by $\tau$, and letting $\tau \to 0$ gives

$$\partial_t\rho_t(\theta) = \alpha \int_\Theta R(\theta, \theta')\left(\rho_*(\theta)\rho_t(\theta') - \rho_t(\theta)\rho_*(\theta')\right)d\theta'. \tag{23}$$

We can now add the Langevin terms that arise in the continuous limit of the compounded MCMC scheme that we use, to arrive at

$$\partial_t\rho_t = \nabla \cdot (\rho_t\nabla U_* + \nabla\rho_t) + \alpha \int_\Theta R(\theta, \theta')\left(\rho_*(\theta)\rho_t(\theta') - \rho_t(\theta)\rho_*(\theta')\right)d\theta' \tag{24}$$

where $\alpha > 0$ measures the separation of time scale between the Langevin and the resampling terms. Written in term of $g_t = \rho_t/\rho_*$ and $\hat{g}_t = \hat{\rho}_t/\rho_*$ (now also allowed to vary with time) Eq. (24) reads

$$\partial_t g_t = -\nabla U_* \cdot \nabla g_t + \Delta g_t + \alpha \int_\Theta \min(\hat{g}_t(\theta), \hat{g}_t(\theta'))\left(g_t(\theta') - g_t(\theta)\right)\rho_*(\theta')d\theta' \tag{25}$$

## B.3. Convergence rate

Consider the Pearson $\chi^2$-divergence of $\rho_t$ with respect to $\rho_*$ defined

$$D_t = \int_\Theta \frac{\rho_t^2}{\rho_*}d\theta - 1 = \int_\Theta g_t^2\rho_*d\theta - 1 \geq 0. \tag{26}$$

Assuming that $D_0 < \infty$ and using (25) we deduce that $D_t$ satisfies

$$\begin{aligned}
\frac{dD_t}{dt} &= 2\int_\Theta g_t(\theta)\partial_t g_t(\theta)\rho_*(\theta)d\theta \\
&= 2\int_\Theta g_t(\theta)\nabla \cdot (\rho_*(\theta)\nabla g_t(\theta))d\theta + 2\alpha \int_{\Theta^2} \min(\hat{g}_t(\theta), \hat{g}_t(\theta'))\left(g_t(\theta') - g_t(\theta)\right)g_t(\theta)\rho_*(\theta)\rho_*(\theta')d\theta d\theta' \\
&= -2\int_\Theta |\nabla g_t(\theta)|^2\rho_*(\theta)d\theta - \alpha \int_{\Theta^2} \min(\hat{g}_t(\theta), \hat{g}_t(\theta'))\left|g_t(\theta') - g_t(\theta)\right|^2\rho_*(\theta)\rho_*(\theta')d\theta d\theta' \\
&\leq -\alpha \int_{\Theta^2} \min(\hat{g}_t(\theta), \hat{g}_t(\theta'))\left|g_t(\theta') - g_t(\theta)\right|^2\rho_*(\theta)\rho_*(\theta')d\theta d\theta'
\end{aligned} \tag{27}$$

where we used $(-\nabla U_* \cdot \nabla g_t + \Delta g_t)\rho_* = \nabla \cdot (\rho_* \nabla g_t)$ to reexpress the first integral in the second equality. If we denote $\hat{G}_t = \inf_{\theta \in \Theta} \hat{g}_t(\theta) \in [0, 1]$, (27) implies

$$\frac{dD_t}{dt} \leq -\alpha \hat{G}_t \int_{\Theta^2} |g_t(\theta') - g_t(\theta)|^2 \rho_*(\theta)\rho_*(\theta')d\theta d\theta' = -2\alpha \hat{G}_t D_t, \tag{28}$$

where we used the normalization conditions $\int_\Theta g_t(\theta)\rho_*(\theta)d\theta = \int_\Theta \hat{\rho}(\theta)d\theta = 1$. As a result, using Gronwall inequality we deduce

$$D_t \leq D_0 e^{-\alpha \int_0^t \hat{G}_s ds}. \tag{29}$$

This equation indicates that $D_t \to 0$ as $t \to \infty$ as long as $\int_0^t \hat{G}_s ds \to \infty$. That is, convergence can only fail if $\hat{G}_t = o(t^{-1})$ as $t \to \infty$, and it is guaranteed otherwise. Convergence is also exponential asymptotically, as long as $\hat{G}_t$ remains bounded away from 0 as $t \to \infty$.

To get a more explicit convergence rate, let us analyze (29) in two subcases. First let us assume that the map is not trained, i.e. $\hat{g}_t(\theta) = \hat{g}(\theta)$ is fixed, and denote $\hat{G} = \inf_{\theta \in \Theta} \hat{g}(\theta) \in [0, 1]$. In this case, (29) reduces to

$$D_t \leq D_0 e^{-2\alpha \hat{G} t} \qquad (\hat{g}_t = \hat{g} \text{ fixed}). \tag{30}$$

Note that this bound is only nontrivial if $\hat{G} > 0$. Even if that is the case, the rate in (30) can be pretty poor if $\hat{G}$ is very small (e.g exponentially small in the input dimension $d$), which is to be expected if the map is not trained. The best case scenario is of course the idealized situation when $\hat{G} = 1$, which requires that $\hat{g} = 1$ (i.e. $\hat{\rho} = \rho_*$) because of the normalization conditions $\int_\Theta \hat{g}(\theta)\rho_*(\theta)d\theta = \int_\Theta \hat{\rho}(\theta)d\theta = 1$: this case is the continuous equivalent of the one step convergence of the discrete MCMC scheme with resampling from $\rho_*$.

Second let us assume that $\hat{g}_t = g_t$, that is the trained distribution instantaneously follows the walkers distribution at all times. In this case, (29) reduces to

$$D_t \leq D_0 e^{-2\alpha \int_0^t G_s ds} \qquad (\hat{g} = g_t), \tag{31}$$

where we denote

$$G_t = \inf_{\theta \in \Theta} \left( \frac{\rho_t(\theta)}{\rho_*(\theta)} \right) = \inf_{\theta \in \Theta} g_t(\theta) \in [0, 1]. \tag{32}$$

To make this bound explicit, let us consider the evolution of $G_t$. Denoting $\theta_t = \operatorname{argmin}_{\theta \in \Theta} g_t(\theta)$ so that $G_t = g_t(\theta_t)$, and using $\min(g_t(\theta_t), g_t(\theta')) = g_t(\theta_t) = G_t$, $\nabla g_t(\theta_t) = 0$, and $\Delta g_t(\theta_t) \geq 0$ by definition of $\theta_t$, from (25) we have

$$\begin{aligned}
\frac{dG_t}{dt} &= \partial_t g_t(\theta_t) + \dot{\theta}_t \cdot \nabla g_t(\theta_t) \\
&= \Delta g_t(\theta_t) + \alpha G_t \int_\Theta (g_t(\theta') - G_t) \rho_*(\theta')d\theta' \\
&= \Delta g_t(\theta_t) + \alpha G_t - \alpha G_t^2 \\
&\geq \alpha G_t - \alpha G_t^2
\end{aligned} \tag{33}$$

where we used again the normalization conditions $\int_\Theta g_t(\theta')\rho_*(\theta')d\theta' = \int_\Theta \rho_*(\theta')d\theta' = 1$. Eq. (33) implies that

$$\frac{1}{G_t - G_t^2} \frac{dG_t}{dt} \geq \alpha \tag{34}$$

which after integration gives

$$\log\left( \frac{G_t(1 - G_0)}{G_0(1 - G_t)} \right) \geq \alpha t \tag{35}$$

This means that we have

$$G_t \geq \frac{G_0}{G_0 + (1 - G_0)e^{-\alpha t}}. \tag{36}$$

Inserting this equation in (31) and performing the integral explicitly gives

$$D_t \leq \frac{D_0}{(G_0(e^{\alpha t} - 1) + 1)^2}. \tag{37}$$

This bound is only nontrivial if $G_0 \in (0, 1]$.