# OpenReview forum: "Efficient Bayesian Sampling Using Normalizing Flows to Assist Markov Chain Monte Carlo Methods"
_ICML.cc/2021/Workshop/INNF — INNF+ 2021 contributedtalk_

### Official Review · Reviewer_BPDd · 2021-06-11

**Rating:** Accept
**Confidence:** 4

**Summary:**

This paper proposes a new method which merges MCMC and normalizing flows. In particular, the flow helps the MCMC generate data which helps train the flow. This is done by alternating between generating a normalizing flow based sample and a MALA sample. Compared to standard reverse-kl training of normalizing flows, the proposed method claims to avoid failure cases such as mode collapse. Experiments are performed on two test tasks: one toy and one with astrophysics applications.

**Justification For Rating:**

The paper presents an interesting new paradigm to the standard reverse kl training method. In particular, the application of normalizing flows is both interesting and is executed well. The writing is clear and the idea has the potential to be very impactful. I also very much liked the detail in several of the experiments. In particular, the the ablations showing that the NF eventually learns a distribution that is more accepted is very interesting.

However, I do have several questions about some parts of the paper.

1. How does the algorithm compare against baselines on the experiments. In particular, I am interested in seeing how it compares to reverse-kl training (in terms of accuracy of modelling the distribution) and standard MCMC techniques (in terms of speed).

2. Does the general idea for the algorithm work for other MCMC methods besides MALA? Were any experiments performed here and, if so, what was the performance?

3. Since the algorithm initially maps through a normalizing flow before continuing, do any of the problems with reverse-kl manifest? In particular, I would expect that early on the flow would get rejected with high probability, inhibiting training. What are the effects of varying the time in which we first go through the normalizing flow?

---

### Official Review · Reviewer_CpaW · 2021-06-11

**Rating:** Accept
**Confidence:** 3

**Summary:**

In MCMC, a common strategy to ensure efficient exploration is to perform a combination of small, "local" mutations and large, "global" mutations.

In this work, the authors propose to train a normalizing flow to perform global mutations approximately proportional to the posterior distribution in Bayesian inference. Local mutations are carried out by an arbitrary existing method (e.g. MALA).

The contribution is a combination of (i) training directly from the KL divergence in a MCMC context (ignoring additive and multiplicative constants) as well as (ii) demonstrating superiority to a prior method in estimating the radial velocity of exoplanets.

**Justification For Rating:**

The paper is well written and technically sound. The contribution is self-contained and appropriate for the workshop. I especially appreciate that the numerical experiment was described in a way that a non-expert in physics was able to understand what's going on.

Just a few minor things that I think should be addressed, though:
- Line 88 should state that p_star>0 implies p^hat>0, but not the other way around.  I.e. it's fine if p_hat>0 in places where p_star=0.
- Training a normalizing flow for Monte Carlo integration by minimizing the direct KL divergence has previously been done in Neural Importance Sampling [Müller et al. 2019]. While the context was outside of Markov chains, the underlying principles of importance sampling are the same, so a citation would be appreciated.

---

### Official Review · Reviewer_ApcK · 2021-06-11

**Rating:** Accept
**Confidence:** 4

**Summary:**

Authors consider the classic problem of Bayesian inference: given the prior distribution, the likelihood function, and some data, we would like to sample from the posterior distribution and/or evaluate its normalizing constant. MCMC methods are a popular solution, but it's difficult to get them to mix well when local transitions are used and/or when the target posterior has disconnected modes.

Recently normalizing flows have emerged as a way to parameterize complex distributions via invertible transformations of a simple base distribution. A normalizing flow could be fit to the posterior, turning the inference problem into an optimization problem. Unfortunately, lacking samples from the posterior, we are only able to use the "reverse" KL ($KL\left[p_{\mathit{flow}}||p_{\mathit{true}}\right]$) to fit the flow in this context. Such a KL is "mode seeking", and is prone to "mode collapse", i.e. the flow is likely to miss some modes in the posterior completely.

In this paper, authors propose a method that a) allows to fit the normalizing flow on (the approximation of) the "forward" KL divergence; and b) learns a "data-informed" proposal distribution to improve the mixing of an MCMC sampler. A normalizing flow is trained to be a non-local, "data-informed" proposal distribution for MCMC. The samples from the MCMC sampler are used as a surrogate of the samples from the posterior to train the normalizing flow via the "forward" KL divergence. As the normalizing flow is getting closer the the true posterior distribution during training, the acceptance rate of the MCMC sampler should improve, and vice versa: more samples from the MCMC lead to a better normalizing flow fit. To kick-start the training process and to facilitate exploration, the additional "local" MCMC sampler is combined with the "global" one, alternating between the steps of the two samplers.

Authors showcase the method via numerical experiments on both a synthetic and a real astronomical dataset.

**Justification For Rating:**

"Using a normalizing flow to learn a better proposal distribution for MCMC" is a very interesting direction which there is surprisingly little work in. In the cited (Albergo et al., 2019) MCMC is only used to "refine" the samples from the pre-trained flow (as far as I can tell); and I am *assuming* (Gabrie et al., 2021) is the extended version of this work. (Be careful about citing your own work like this --- reviewers with less good will might consider the submitted paper to be a re-hashing of someone else's work.)

One of the motivations of the method is to be able to use the "forward" KL divergence to avoid "mode collapse" when fitting normalizing flows to Bayesian posteriors. However, the proposed method relies on all the modes or "basins of interest" being provided as "prior knowledge". This is understandable, as the local MCMC sampler is unlikely to discover disconnected modes by itself, and the normalizing flow proposal won't help with mode discovery. Running initial optimization to find the modes as done by the authors should help, but clearly isn't fool-proof. This sounds like an important issue in practical applications. On the other hand, if we *do* know the locations of the modes in advance and want to learn the full density (or, say, the relative weights of the modes), the method gives us a principled way to do that.

The other motivation is to improve the mixing of MCMC. My interpretation of why the proposed method might be helpful is that the normalizing flow effectively represents the "history" of density-weighted locations visited by the local sampler, which we also explicitly populate with the prior knowledge of the mode locations before training. This allows us to use a second sampler with the normalizing flow proposal to periodically make longer jumps to the previously explored modes. To this end, I believe it is important to compare and contrast the method with other "mode-jumping" MCMC methods, such as [1] and [2], which also aim to improve sampling efficiency when the modes are known in advance.

The paper is written well: the exposition is good, the proposed method and its components are explained with great rigour, and experiments are chosen well to showcase the resulting behaviour (as much as it could be given the short page limit).

I believe the work will be of interest to the workshop participants, and will stimulate discussion around learning density models to be used as proposals for MCMC samplers, and hence I recommend accepting the paper. For the future versions of the work I encourage the authors to pay special attention to the potential issues above, and to compare the method to stronger baselines.

Minor:
- Typo in conclusion: "similar to that used *in* (?) variational inference".
- In introduction: replace "the likelihood of the data" with "the likelihood of the parameters $\theta$ given the data". Likelihood $L\left( \\theta\right)$ is defined as the *probability of the data*, given the parameters. The likelihood function itself is *not* a probability distribution, so it's useful to make a distinction between the two. See top of page 29 in David MacKay's book for more context.
- Please rasterize the plots in Figure 3: my PDF reader struggles to render all  the points while scrolling. You can pass "rasterized=True" to scatter() to do so in matplotlib. Future readers will thank you. :-)

*References:*

[1]: Tjelmeland, H. and Hegstad, B.K. (2001), Mode Jumping Proposals in MCMC. Scandinavian Journal of Statistics, 28: 205-223. https://doi.org/10.1111/1467-9469.00232

[2]: Sminchisescu, C. and Welling, M. (2007), Generalized Darting Monte Carlo. Artificial Intelligence and Statistics, PMLR, 516–23. http://proceedings.mlr.press/v2/sminchisescu07a.html.

---

### Decision · Program_Chairs · 2021-06-14

Accept (contributed talk)